# 3D Shape Generation via Tri-Vectors: a Parsimonious Representation

## Abstract

The pursuit of optimal 3D representations remains both a long-standing challenge and an exciting frontier within the vision and graphics communities. We argue that compressing 3D data into low-dimensional components improves parameter efficiency and captures essential features, providing a parsimonious representation that enhances shape generation. To this end, we propose Tri-Vectors, a parsimonious 3D representation tailored for shape generation. Tri-Vectors instantiates the classical CANDECOMP/PARAFAC (CP) decomposition of a shape's continuous signed distance field (SDF) into orthogonal tri-vector sets, yielding a compact, resolution-independent, and highly adaptable structure. Specifically, Tri-Vectors has three major advantages: (i) direct shape reconstruction through linear combinations of components, (ii) adjustable dimension and number of components to suit varying shape complexities, and (iii) robustness across arbitrary resolutions. Extensive experiments across multiple datasets show that Tri-Vectors outperforms state-of-the-art methods in terms of parameter efficiency and geometric fidelity. Moreover, we extend its application to textured and deformable shapes, demonstrating the scalability and versatility of the representation.

## 1 Introduction

> *"Entities should not be multiplied unnecessarily."*
>
> *– William of Ockham*

With the rapid growth of virtual and augmented worlds, the demand for high-quality 3D assets has increased dramatically. Efficiently generating 3D models is, however, substantially more challenging than learning in 2D. This is due not only to the higher ambient dimensionality but also to the irregular and complex topology of 3D shapes, leading to increased memory traffic and computational demands. As curated 3D datasets continue to expand, a central question emerges: *What makes an ideal 3D representation for generative tasks?* The choice of representation serves as the interface between geometry and the generator, directly affecting parameter count, memory bandwidth, and how gradient signals capture geometric structure. Yet, despite significant progress (Kang et al., 2025; Zhang et al., 2024b; Lai et al., 2025; Chen et al., 2025; Xiang et al., 2025; Wu et al., 2025), existing representations face fundamental trade-offs between expressiveness, efficiency, and generalization.

Neural implicit functions have demonstrated strong ability to represent diverse shapes (Zhang et al., 2024b; 2023), but capturing fine-grained details often requires high-capacity networks, resulting in heavy compute and memory footprints. Beyond implicit fields, recent works explore tensor-structured parameterizations for shapes or scenes (Shue et al., 2023; Gao et al., 2023; Chen et al., 2022; 2023; Hui et al., 2024; Hu et al., 2024; Hui et al., 2022), but are inherently scene-specific and often rely on auxiliary neural decoders, limiting out-of-distribution generalization and scalability for shape generation. These limitations point to a persistent gap: current approaches fail to uncover and leverage the intrinsic low-dimensional structures of 3D data. This motivates our central research question: *"How can we design a parsimonious 3D representation that is both parameter-efficient and structurally simple, while retaining adaptability and fidelity for shape generation?"*

Among classical tensor factorizations, the CP decomposition (Carroll & Chang, 1970) provides a parsimonious way to represent a 3D tensor as a sum of separable rank-one components. CP has been widely studied in mathematics, signal processing, and scientific computing. Moreover, CP-style factorizations have been adopted to parameterize shapes or scenes for reconstruction and rendering (Chen et al., 2022; Gao et al., 2023); however, their potential as a representation for 3D shape generation has not been explored. This gap motivates us to revisit CP in a generative setting.

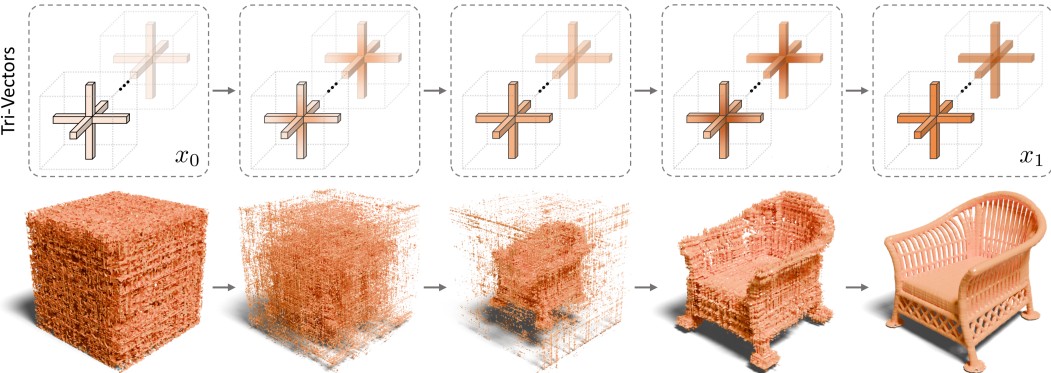

Figure 1: 3D shape generation process based on our Tri-Vectors representation using Flow-Matching model. *Top:* illustrates the generation process applied to Tri-Vectors. *Bottom:* displays the corresponding 3D volumes reconstructed from tri-vectors at each timestep.

We present Tri-Vectors, a simple yet effective representation that brings CP decomposition to 3D shape generation, (see Fig. 1). It decomposes the continuous SDF into multiple orthogonal tri-vector sets, addressing key limitations in existing methods. Tri-Vectors offers three essential properties: **(1) Efficiency:** Tri-Vectors reconstruct shapes via a linear combination of structured components without auxiliary neural decoder, enabling a compact and efficient shape representation. **(2) Adaptability:** The vector dimensionality and component count are adjustable, allowing flexible adaptation to shapes of varying complexity and enhancing generalization across tasks. **(3) Scalability:** Unlike grid-based methods, Tri-Vectors are resolution-independent, supporting reconstruction at arbitrary detail levels while maintaining geometric fidelity.

We conduct extensive experiments across multiple datasets, covering 3D reconstruction and generation tasks of varying complexity. The results reveal that Tri-Vectors compresses shape information effectively and outperforming state-of-the-art methods in terms of parameter efficiency and geometric fidelity. Moreover, we explore the potential of Tri-Vectors for textured and deformable shapes, decomposing color spaces for textures and incorporating a temporal dimension to represent 4D deformable shapes. This further demonstrates its scalability and adaptability, making Tri-Vectors a promising foundation for future 3D generative applications.

## 2 RELATED WORKS

### 2.1 3D SHAPE REPRESENTATIONS

Accurate and efficient representation of 3D data has been a longstanding challenge. *Point cloud* (Chen et al., 2024a; Qi et al., 2017a;b; Nichol et al., 2022) captures spatial positions but lacks connectivity, complicating surface inference. *Mesh* (Alliegro et al., 2023; Chen et al., 2024b; Siddiqui et al., 2024; Chen et al., 2024c; Nash et al., 2020) provides detailed topology but is difficult for neural networks to process due to structural irregularities. *Voxel grid* (Wu et al., 2015; 2016; Zheng et al., 2022; Wang et al., 2024; Ren et al., 2024) offers regularity for convolutions but memory-intensive, limiting scalability. *Octree* (Wang et al., 2017; 2022; Zheng et al., 2023) uses hierarchical partitioning to allocate resources adaptively, but introduces additional computational complexity. *Neural field* (Park et al., 2019; Mescheder et al., 2019; Chen & Zhang, 2019; Erkoç et al., 2023; Zhang et al., 2022; 2023) represents shapes as continuous functions, offering compactness but often requires high-capacity networks or auxiliary encoders for detail.

Recent advances in *hybrid representations* (Shue et al., 2023; Hui et al., 2022; Chen et al., 2022; 2023; Müller et al., 2022; Gao et al., 2023; Yariv et al., 2024), summarized in Tab. 1, try to balance efficiency and expressiveness across applications. Methods such as TensoRF (Chen et al., 2022), DictionaryFields (Chen et al., 2023), and StriVec (Gao et al., 2023) decompose scenes into multiple tensor components, enabling fast querying and rendering. However, these methods are inherently designed for scene-specific tasks and, despite being applicable to SDF representation, are *not intended for generalizable shape generation*. Additionally, when applied to shape modeling via SDF, they introduce significant parameter redundancy. For shape generation, Triplanes (Shue et al., 2023) encodes shapes using tri-planes but highly relies on an additional MLP decoder, increasing computational complexity and restricting generalization. NeuralWavelet (Hui et al., 2022) applies wavelet decomposition to fixed-resolution SDF grids, limiting adaptability to continuous spaces and high-

| Method | Generation | MLP-free | Scalability |
|---|---|---|---|
| TensoRF (Chen et al., 2022) | ✗ | ✗ | ✓ |
| InstantNGP (Müller et al., 2022) | ✗ | ✗ | ✓ |
| DictionaryFields (Chen et al., 2023) | ✗ | ✗ | ✓ |
| StriVec (Gao et al., 2023) | ✗ | ✓ | ✓ |
| Tri-plane (Shue et al., 2023) | ✓ | ✗ | ✓ |
| NeuralWavelet (Hui et al., 2022) | ✓ | ✓ | ✗ |
| MSDF (Yariv et al., 2024) | ✓ | ✓ | ✗ |
| **Ours** | ✓ | ✓ | ✓ |

Table 1: Comparison of different 3D representations in terms of their suitability for shape generation, reliance on MLP-based decoding, and scalability (resolution-independent).

resolution modeling. MSDF (Yariv et al., 2024) builds local SDF grids around sampled surface points, with accuracy dependent on sampling density and grid resolution, and requiring careful design for global consistency. Ultimately, both NeuralWavelet (Hui et al., 2022) and MSDF (Yariv et al., 2024) remain discretized representations of SDFs, where their expressiveness is inherently constrained by predefined grid resolution or sampling density. This fundamental problem reinforces the challenge of balancing structural efficiency with representational flexibility.

## 2.2 3D SHAPE GENERATION

The development of diverse 3D datasets (Chang et al., 2015; Collins et al., 2022; Zhou & Jacobson, 2016; Deitke et al., 2023b;a) has enabled researchers to generate high-fidelity 3D assets, driving the continuous innovation of 3D generative models.

Transformer-based auto-regressive models generate 3D assets by modeling sequential dependencies (Yan et al., 2022; Nash et al., 2020; Jayaraman et al., 2023; Sun et al., 2020; Medi et al., 2023; Ma et al., 2025; Zhang et al., 2024a). Decoder-only transformers predict triangle meshes, with some incorporating VQ-VAE for discrete encoding (Siddiqui et al., 2024; Chen et al., 2024b;c). However, token length limitations necessitate mesh simplification or extra processing, restricting complex scene generation. SDF-based approaches partition SDFs into grids and use progressive prediction but require additional models for SDF encoding (Mittal et al., 2022; Yan et al., 2022). Diffusion models (Ho et al., 2020) have demonstrated strong potential in 3D generation, being applied to point clouds (Luo & Hu, 2021), voxel grids (Hu et al., 2024; Hui et al., 2022; 2024; Wang et al., 2024; Chou et al., 2023; Zheng et al., 2023), polygon meshes (Alliegro et al., 2023), and neural fields (Chen & Zhang, 2019; Erkoç et al., 2023). However, the high dimensionality of 3D data poses challenges in efficiency and quality. Inspired by latent diffusion models (Rombach et al., 2022), recent works train diffusion models in the latent space using VAE-encoded 3D representations (Zhang et al., 2023; Xiong et al., 2024; Ren et al., 2024; Zhang et al., 2024b; Wu et al., 2024; Cui et al., 2024; Gupta et al., 2023; Zhang et al., 2022).

As 3D generative models continue to evolve, the emergence of more compact and expressive representations plays a key role in improving efficiency, fidelity, and scalability (Hui et al., 2024; Yariv et al., 2024; Wu et al., 2024; Lee et al., 2024; Zhang et al., 2024b). By instantiating a CP decomposition, Tri-Vectors achieves high geometric fidelity with significantly fewer parameters due to its inherent compactness. Its simple, structured design integrates seamlessly with neural networks, ensuring efficient processing and accurate learning.

## 3 METHOD

We aim to represent redundant SDFs as parsimonious Tri-Vectors. A key observation is that most geometric information in 3D shapes concentrates near the surface, while off-surface regions contribute little. However, SDF ignores this pattern, encoding shapes as dense, continuous volumetric fields, leading to redundancy. Our Tri-Vectors eliminate this inefficiency by leveraging CP decomposition to efficiently factorize SDFs, preserving their smooth global structure while concentrating parameters on localized details, resulting in a compact yet expressive representation. The method overview is shown in Fig. 2.

Figure 2: Overview of our approach. (a): *Shape Learning* introduces how Tri-Vectors decompose the shape's truncated signed distance field (TSDF) into multiple orthogonal tri-vector sets. (b): *Training* shows how to train a flow-based generative model to produce Tri-Vectors from a random noise sampled from Gaussian distribution. (c): *Shape Generation* employs the trained model to generate a Tri-Vectors and then reconstruct a dense SDF grid with an arbitrary resolution, flowed by marching cube (Lorensen & Cline, 1987) we can generate the output 3D shape.

### 3.1 TRI-VECTORS REPRESENTATION

Our Tri-Vectors approximates the original dense continuous SDF by expressing it as a sum of discrete, orthogonal tri-vector sets. This results in a significantly more compact representation while preserving the key geometric features of the original shape. Mathematically, the original CP decomposition of a 3D tensor $\mathcal{T} \in \mathbb{R}^3$ is given by Eq. 1.

$$\mathcal{T} \approx \sum_{r=1}^{R} \mathbf{a}_r \otimes \mathbf{b}_r \otimes \mathbf{c}_r, \tag{1}$$

where $\mathbf{a}_r$, $\mathbf{b}_r$, and $\mathbf{c}_r$ are one-dimensional vectors that encode the information along each axis, and $\otimes$ denotes the out product. By adjusting the *rank $R$* (the number of components in the summation) and the *resolution* of each vector, we can effectively control the balance between compression rate and reconstruction fidelity. This approach reduces the memory footprint of the shape representation, making it feasible to store complex shapes.

**Shape learning.** Given an input 3D shape $\mathcal{S}$, our goal is to compute its Tri-Vectors representation: $\mathcal{S}_{\text{TV}} = \{(\mathbf{x}_1, \mathbf{y}_1, \mathbf{z}_1), \ldots, (\mathbf{x}_R, \mathbf{y}_R, \mathbf{z}_R)\}$. Fig. 2 (a) illustrates the process of convert a shape into Tri-Vectors. We start by normalizing the shape within a unit cube $(-0.5, 0.5)$. To optimize the Tri-Vectors, we sample 6 million points: 2 million points are uniformly distributed throughout the cube, and 4 million points are concentrated near the shape's surface to capture finer geometric details, of these, 2 million are sampled directly on the surface, while the other 2 million are obtained by adding uniformly distributed noise in the range $[-0.01, 0.01]$ to the surface points. For each sampled point $p_n$, we compute the ground-truth SDF value $s(p_n)$ and truncate them to $[-0.05, 0.05]$. The Tri-Vectors approximates the truncated SDF value at any point $p_n$ by combining the contributions from each tri-vector as follows:

$$\hat{s}(p_n) = \sum_{r=1}^{R} \mathbf{x}_{r,i} \cdot \mathbf{y}_{r,j} \cdot \mathbf{z}_{r,k}, \tag{2}$$

where $\mathbf{x}_{r,i}$, $\mathbf{y}_{r,j}$, and $\mathbf{z}_{r,k}$ are the elements of the vectors corresponding to the coordinates of $p_n = (i, j, k)$ in the 3D space. For any point in the space, we use linear interpolation on each vector to obtain the corresponding value.

To optimize this representation, we minimize the distance between the reconstructed values $\hat{s}(p_n)$ and the ground-truth truncated SDF values $s(p_n)$. The objective function is defined as an $L_2$ loss:

$$\mathcal{L}_{\text{recon}} = \sum_{n=1}^{N} \|\hat{s}(p_n) - s(p_n)\|_2^2, \tag{3}$$

where $N$ is the total number of sample points.

Additionally, we include $L_2$ regularization term with weight $\lambda_{\text{L2}}$ to discourage outlying values and total variation regularization term (Shue et al., 2023) with weight $\lambda_{\text{var}}$ (shown Eq. 4) to make the

representation become smoother:

$$\mathcal{L}_{\text{var}} = \left( \sqrt{\sum (\Delta_e^2)} + \sqrt{\sum (\Delta_c^2)} \right), \tag{4}$$

where $\Delta_e$ represents the differences between adjacent elements within each vector, and $\Delta_c$ denotes the differences between corresponding elements of adjacent components.

The final loss function combines these components to balance shape reconstruction accuracy and smoothness:

$$\mathcal{L}_{\text{total}} = \mathcal{L}_{\text{recon}} + \lambda_{\text{L2}} \cdot \mathcal{L}_{\text{L2}} + \lambda_{\text{var}} \cdot \mathcal{L}_{\text{var}}, \tag{5}$$

in our experiments, $\lambda_{\text{L2}} = 1\text{e-}10$ and $\lambda_{\text{var}} = 1\text{e-}8$.

The Tri-Vectors seeks the optimal vectors $\{\mathbf{x}_r, \mathbf{y}_r, \mathbf{z}_r\}_{r=1}^R$ that minimize this total loss, which can be formulated as:

$$\{\mathbf{x}_r, \mathbf{y}_r, \mathbf{z}_r\} = \arg \min_{\{\mathbf{x}_r, \mathbf{y}_r, \mathbf{z}_r\}} \mathcal{L}_{\text{total}}. \tag{6}$$

By iteratively optimizing these factors using gradient descent, we obtain a compact Tri-Vectors that can accurately approximate a shape. This method enables efficient storage and reconstruction of the shape, capturing essential geometric details with high fidelity, without the need for any neural networks. Fig. 3 illustrates the progressive improvement in reconstruction quality as both *resolution* and *rank* increase, demonstrating the trade-off between parameter count and reconstruction fidelity.

## 3.2 SHAPE GENERATION

Tri-Vectors' simplicity and highly structured form make it ideal for seamless integration into modern neural networks, especially transformer-based architectures, without additional processing. Specifically, we transform *3D shape generation* into *1D sequence generation*, which can not only capture essential geometric information with much fewer parameters but also transform spatial dependencies into sequential dependencies, rather than relying on multiple convolutions to obtain a large receptive field. By doing so, it reduces training complexity while still accurately modeling long-range dependencies, making the generative process more effective and scalable.

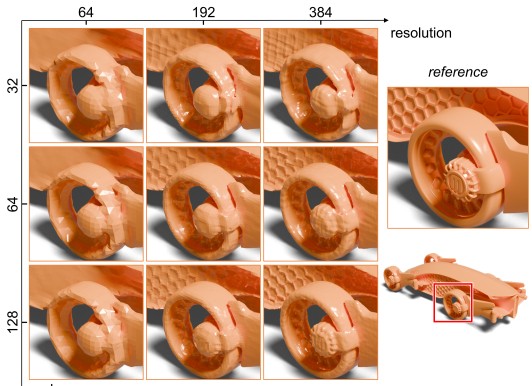

Figure 3: Shape reconstruction from Tri-Vectors with different configurations.

To instantiate this idea, we introduce SiT (Ma et al., 2024) as our base generative model. SiT proposes an interpolant process $I(t)$ that smoothly transforms noise to data over time $t$. This process is governed by a transform function $T_t(x)$, where $T_t : x \rightarrow$ data for $t \in [0, 1]$, bridging flow and diffusion models. In this case, Tri-Vectors can be seamlessly integrated into SiT (see Fig. 2), using its powerful modeling capabilities for shape generation. We also adopt score estimation $\nabla \log p(x)$ for data distribution learning, and leverage transformers to capture dependencies across time steps.

Moreover, to enhance the network's learning capabilities, we introduce specialized positional embeddings designed for Tri-Vectors: (1) *axis embeddings* that indicate whether the input token corresponds to the $x$, $y$, or $z$ axis; (2) *component embeddings* that identify the specific component within the sequence; and (3) *sequence position embeddings* that mark the position of each token within the overall sequence. Note that all embeddings in this framework are learnable, allowing for adaptive learning that accelerates convergence and improves final performance.

## 4 EXPERIMENTS

Through our comprehensive experiments, we aim to answer four central questions regarding the effectiveness of Tri-Vectors. **Q1**: Can Tri-Vectors achieve greater efficiency and parsimony in geometric representation compared to existing approaches? **Q2**: As representation capacity increases,

how do Tri-Vectors balance expressiveness with parameter efficiency? **Q3**: What advantages does the parsimonious nature of Tri-Vectors bring to practical 3D generative tasks? **Q4**: Do Tri-Vectors exhibit extensibility and scalability, enabling adaptation to more complex scenarios such as textured and deformable shapes?

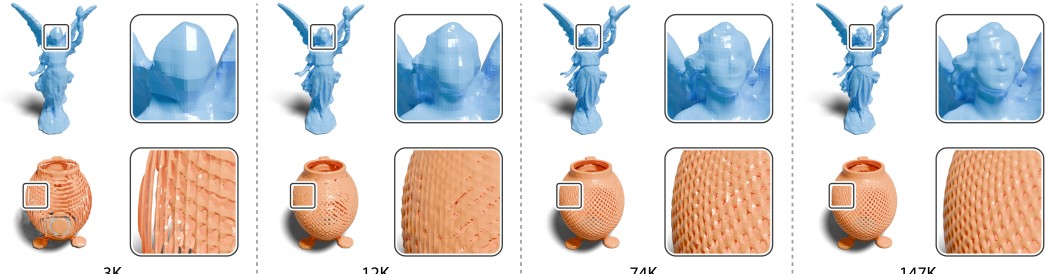

3K 12K 74K 147K

Figure 4: Reconstruction results from Tri-Vectors with different parameter counts (3K: rank=16, resolution=64; 12K: rank=32, resolution=128; 74K: rank=128, resolution=192; 147K: rank=128, resolution=384), showing improved detail capture as parameter capacity scales. Our method follows a scaling law that enhances expressiveness while maintaining parsimonious use of parameters.

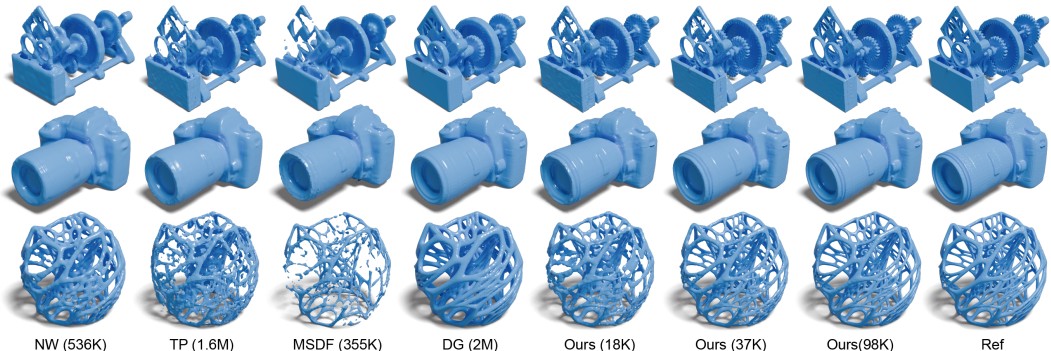

NW (536K)  TP (1.6M)  MSDF (355K)  DG (2M)  Ours (18K)  Ours (37K)  Ours(98K)  Ref

Figure 5: Shape reconstruction from Tri-Vectors with varying parameter counts (18K: rank=32, resolution=192; 37K: rank=64, resolution=192; 98K: rank=128, resolution=256) compared to other representations: NW, TP, MSDF and Dense Grid.

### 4.1 SHAPE LEARNING

To answer **Q1** and **Q2**, we first evaluate the performance of Tri-Vectors in compression and reconstruction in different parameter configurations, then, we compare it to existing SDF based representations commonly used in 3D generative models such as dense 3D volumetric grids at 128 spatial resolution (DG), Triplane (Shue et al., 2023) (TP), NeuralWavelet (Hu et al., 2024) (NW), and MSDF (Yariv et al., 2024) (MSDF). Furthermore, we compare our method against methods designed for fast query and rendering, including StriVec (Gao et al., 2023), InstantNGP (Müller et al., 2022) (NGP), and DictionaryFields (Chen et al., 2023) (DF). Please refer to the supplementary materials for more experimental details.

**Results and Discussion.** Fig. 4 illustrates the Tri-Vectors representing shapes with different parameter counts. Even with only 3K parameters, Tri-Vectors captures the global geometric structure well. As the parameter count increases, the representation progressively reveals finer details, demonstrating a smooth and efficient scaling in expressiveness and robustness. This because a shape's SDF naturally exhibits a *globally smooth with localized complexity structure*, which aligns well with the strength of CP decomposition in capturing separable structures. Thus, by instantiating CP decomposition, Tri-Vectors achieves efficient compression while preserving geometric detail.

Fig. 6 displays that Tri-Vectors preserve global shape structure under partial information loss: randomly masking subsets of tri-vector components leaves the coarse geometry intact while only mildly degrading fine-scale detail. This resilience improves training stability and enables high-quality reconstructions even with noisy or incomplete inputs.

| Model | #Param. | Thingi10K | | | | ShapeNet | | | |
|---|---|---|---|---|---|---|---|---|---|
| | | CD ($\downarrow$) | IoU ($\uparrow$) | F-Score ($\uparrow$) | Time ($\downarrow$) | CD ($\downarrow$) | IoU ($\uparrow$) | F-Score ($\uparrow$) | Time ($\downarrow$) |
| MSDF(Yariv et al., 2024) | 355K | 12.4 | 0.9573 | 0.61 | 120 s | 10.1 | 0.9950 | 0.73 | 120 s |
| NW(Hui et al., 2022) | 536K | 10.2 | 0.9681 | 0.65 | – | 6.67 | 0.9986 | 0.84 | – |
| TP(Shue et al., 2023) | 1.6M | 15.9 | 0.9934 | 0.53 | 100 s | 13.7 | 0.9901 | 0.56 | 100 s |
| DG($128^3$) | 2M | 15.1 | 0.9914 | 0.31 | – | 17.9 | 0.9933 | 0.48 | – |
| Ours | 18K | 6.09 | 0.9982 | 0.77 | **30 s** | 4.89 | 0.9991 | 0.88 | **30 s** |
| | 37K | 5.52 | 0.9989 | 0.81 | 60 s | 4.93 | 0.9992 | 0.88 | 60 s |
| | 98K | **5.25** | **0.9994** | **0.83** | 80 s | **4.55** | **0.9995** | **0.90** | 80 s |

Table 2: Reconstruction performance from representation of Tri-Vectors compared to other methods on Thingi10K and ShapeNet datasets. Note the Tri-Vectors are evaluated with three different parameter counts, and CD is reported in units of $10^{-3}$.

Tab. 2 shows the reconstruction performance of different methods. Our Tri-Vectors consistently outperforms other methods on both simple and complex shapes with much fewer parameters, such as 18K, verifying its parsimony in geometric representation. Among the methods that require optimization, Tri-Vectors also achieves the best performance in terms of optimization time. The visual comparisons in Fig. 5 further confirm that Tri-Vectors captures fine details more effectively than competing approaches, even with a compact parameter budget.

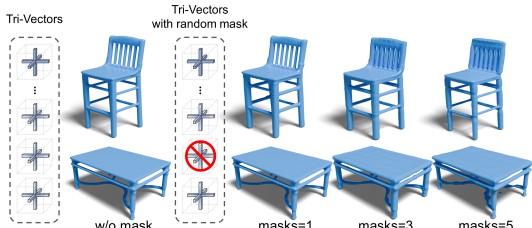

Figure 6: Shape reconstruction from Tri-Vectors with random mask. We randomly masking different numbers of tri-vector of shape's Tri-Vectors.

| Model | CD ($\downarrow$) | IoU ($\uparrow$) | F-Score ($\uparrow$) |
|---|---|---|---|
| NGP (Müller et al., 2022) | **5.98** | **0.9987** | **0.81** |
| DF (Chen et al., 2023) | 6.04 | 0.9983 | 0.77 |
| StriVec (Gao et al., 2023) | 6.77 | 0.9965 | 0.75 |
| Ours | 6.28 | 0.9981 | 0.77 |

(a) Metrics comparison.

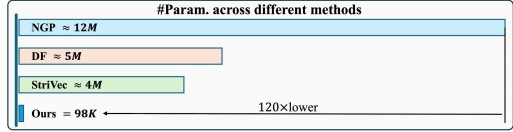

(b) Parameter comparison.

Figure 7: Quantitative reconstruction results and parameter comparison on Thingi10K. (a) While NGP achieves the best metrics, it requires over 12M parameters. In contrast, Tri-Vectors attain comparable reconstruction quality with only 98K parameters—more than 100× fewer than NGP and significantly lower than DF and StriVec. (b) This highlights the superior parameter efficiency of Tri-Vectors without sacrificing geometric fidelity.

Fig. 7 further shows the reconstruction comparison results across different methods, these methods are designed for fast querying and rendering within a single scene, making them unsuitable for generative tasks. In addition, they require significantly more parameters to represent scenes, leading to redundancy in the context of SDF compression. In contrast, Tri-Vectors achieves comparable reconstruction quality while leveraging the intrinsic structure of SDFs to eliminate redundancy. Its ability to capture global structure with localized detail aligns well with global CP decomposition, facilitating compact yet expressive representations.

## 4.2 SHAPE GENERATION

For **Q3**, we train the 3D shape generative model using Tri-Vectors on ShapeNet (Chang et al., 2015) dataset, focusing on four primary categories: *airplane*, *car*, *chair* and *desk* which containing more than 4000 shapes each. Please refer to the supplementary materials for details of implementation.

**Baselines.** We compare our approach with TP (Shue et al., 2023), NW (Hui et al., 2022), and S2VS (Zhang et al., 2023). Both TP and NW compress shape's SDF or occupancy information and transform it into another domain for generative model training, making them the most comparable to our Tri-Vectors. S2VS, on the other hand, employs a VAE to encode shape's occupancy into a latent space before training a generative model.

Figure 8: Comparisons with state-of-the-art methods. From top to bottom: TP (Shue et al., 2023), NW (Hui et al., 2022), S2VS (Zhang et al., 2023), and our Tri-Vectors (18K: rank=32, resolution=192).

| Model | Airplane | | | | | | Car | | | | | |
|---|---|---|---|---|---|---|---|---|---|---|---|---|
| | COV (%, ↑) | | MMD (↓) | | 1-NN (%) | | COV (%, ↑) | | MMD (↓) | | 1-NN (%) | |
| | CD | EMD | CD | EMD | CD | EMD | CD | EMD | CD | EMD | CD | EMD |
| TP (Shue et al., 2023) | 50.12 | 50.03 | 2.83 | 3.07 | 69.51 | 71.81 | 21.47 | 23.79 | 3.81 | 3.07 | 73.07 | 81.70 |
| NW (Hui et al., 2022) | 47.90 | 46.67 | 2.67 | 2.53 | 70.98 | 69.89 | - | - | - | - | - | - |
| S2VS (Zhang et al., 2023) | 48.15 | 51.11 | 2.52 | 2.61 | 70.86 | 66.30 | **45.26** | 44.91 | **2.36** | **2.49** | 86.73 | 80.17 |
| Ours | **55.60** | **52.81** | **2.49** | **2.31** | **60.74** | **64.12** | 43.13 | **45.81** | 2.77 | 2.52 | **70.19** | **77.37** |

| Model | Chair | | | | | | Table | | | | | |
|---|---|---|---|---|---|---|---|---|---|---|---|---|
| | COV (%, ↑) | | MMD (↓) | | 1-NN (%) | | COV (%, ↑) | | MMD (↓) | | 1-NN (%) | |
| | CD | EMD | CD | EMD | CD | EMD | CD | EMD | CD | EMD | CD | EMD |
| TP (Shue et al., 2023) | 43.97 | 41.97 | 7.83 | 5.17 | 59.27 | 63.37 | - | - | - | - | - | - |
| NW (Hui et al., 2022) | 46.31 | 46.09 | 7.60 | 4.56 | 61.50 | 61.52 | 48.29 | 49.80 | 6.45 | 4.11 | 55.05 | 54.68 |
| S2VS (Zhang et al., 2023) | 53.24 | 50.01 | 7.48 | 4.73 | 58.78 | 59.38 | 54.64 | 53.71 | 6.37 | 4.25 | **53.47** | 57.32 |
| Ours | **53.38** | **54.29** | **7.21** | **4.37** | **51.86** | **53.17** | **56.05** | **56.44** | **5.35** | **3.97** | 54.67 | **52.66** |

Table 3: Comparison of models based on COV, MMD, and 1-NN metrics across different categories (Airplane, Car, Chair, and Table).

**Results and Discussion.** Tab. 3 presents the quantitative comparison of COV, MMD, and 1-NNA metrics against baselines. Our Tri-Vectors generative model demonstrates better performance, achieving the best results across most metrics. Fig. 8 provides a qualitative comparison for two representative classes *chairs* and *airplanes* that are shared by all baselines. The results reveal that Tri-Vectors generate shapes with higher fidelity and better-preserved details. This performance is attributed to Tri-Vectors' ability to compress shapes into a compact representation, capturing essential features with fewer parameters, making it easier for the model to learn data patterns and improve generation quality. Moreover, the simple structure of Tri-Vectors enables the use of advanced Transformer-based generative networks, further enhancing the quality of shape generation.

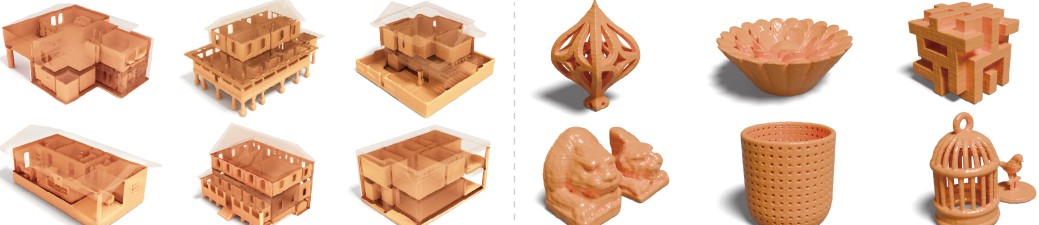

Figure 9: Gallery of generated models showcasing complex structures, each represented by Tri-Vectors (37K: rank=64, resolution=192). *Left:* samples generated from models trained on the BuildingNet dataset. *Right:* samples generated from models trained on the Thingi10K dataset.

Additionally, to further validate the efficiency of Tri-Vectors in representing complex shapes, we curated approximately 1000 samples from both the BuildingNet (Selvaraju et al., 2021) and

Thingi10K, converting each into the Tri-Vectors format. We then trained separate generative models on these datasets. Fig. 9 presents samples generated from the trained models, demonstrating Tri-Vectors' capability to effectively capture and represent intricate geometric details, handling complex models with diverse internal and external structures.

**Shape Novelty Analysis.** Here, we evaluate whether our method can generate shapes beyond merely memorizing the training set. To verify this point, we generated 512 random shapes and retrieved the top three most similar shapes from the training set using Chamfer Distance (CD). Fig. 10 illustrates examples of our generated shapes (orange) alongside their closest matches from the training set (blue). While the generated shapes share overall structural similarities with their nearest counterparts, they also exhibit distinct variations in local features, demonstrating our method's ability to produce novel and realistic structures. More details can be found in supplementary.

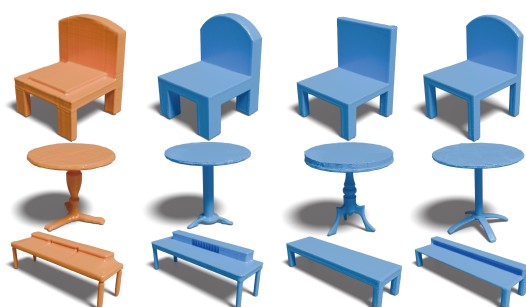

Figure 10: Generated shape novelty analysis. For each generated shape (orange), we retrieve top three most similar shapes (blue) in training set by Chamfer Distance.

### 4.3 MORE APPLICATIONS

Apart from geometric compression, our Tri-Vectors can be extended to handle textured and deformable shapes, demonstrating its adaptability and scalability across diverse 3D tasks. Fig. 11 showcases examples of textured and deformable shape reconstructions using Tri-Vectors. This part answers **Q4** and more details and results are included in the supplementary materials.

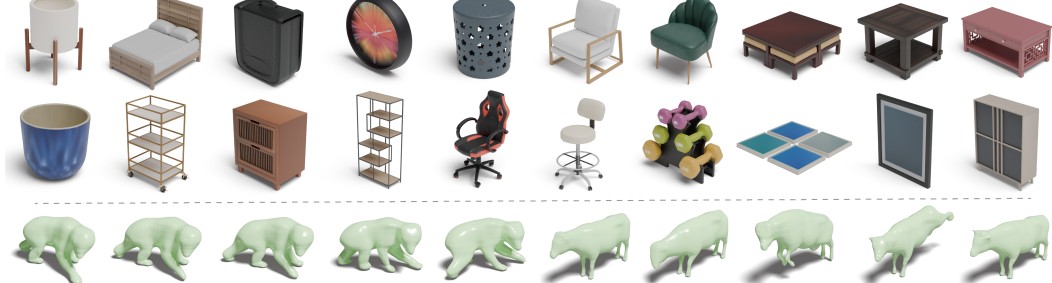

Figure 11: Gallery of applications using Tri-Vectors representation. *Top:* textured shapes reconstruction with 92K parameters. *Bottom:* deformable shapes reconstruction with 98K parameters.

**Tri-Vectors for Textured Shapes.** We represent textured shapes from (Collins et al., 2022) using a multi-component Tri-Vectors formulation, defined as $\mathcal{S}_{\text{TVC}} = \{\mathcal{S}_{\text{TV}}, \mathcal{S}_{\text{R}}, \mathcal{S}_{\text{G}}, \mathcal{S}_{\text{B}}\}$. Here, $\mathcal{S}_{\text{TV}}$ encodes the geometry, while $\mathcal{S}_{\text{R}}$, $\mathcal{S}_{\text{G}}$, and $\mathcal{S}_{\text{B}}$ independently represent the color fields for the RGB channels. By querying the color fields at the vertex positions, we obtain per-vertex RGB values and thus recover the textured geometry.

**Tri-Vectors for Deformable Shapes.** We incorporate an additional time vector for each component, expressing deformable shapes from (Li et al., 2021) as $f(x, y, z, t) = \text{SDF}$. This extension preserves compactness while facilitating smooth interpolation over time, enabling seamless shape transitions.

## 5 CONCLUSION

In this work, we introduce Tri-Vectors, a parsimonious 3D representation that instantiates CP decomposition for shape generation. By adhering to the principles of parameter efficiency and structural simplicity, Tri-Vectors provides a resolution-independent and highly adaptable approach for 3D shape representation, making it easy to integrate with modern generative framework. Across reconstruction and generation benchmarks, Tri-Vectors achieves higher geometric fidelity with substantially fewer parameters than prior methods. We also extend its applicability to textured and deformable shapes, revealing its scalability and versatility in diverse downstream tasks.

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
