# OpenReview forum: "3D Shape Generation via Tri-Vectors: a Parsimonious Representation"
_ICLR.cc/2026/Conference — Submitted to ICLR 2026_

### Official Review · Reviewer_MHCt · 2025-10-27

**Soundness:** 3
**Presentation:** 3
**Contribution:** 2
**Rating:** 6
**Confidence:** 3

**Summary:**

Following the classical CANDECOMP/PARAFAC (CP) decomposition, this work explores tri-vectors of a shape’s signed distance field (SDF) as an alternative continuous shape representation, demonstrating their efficiency in both shape reconstruction and generation.
However, establishing such representations for all shapes in a large dataset, such as ShapeNet, would require substantial time.

**Strengths:**

1. The tri-vectors represent surfaces in a continuous manner while achieving higher parameter efficiency compared to SDFs.
2. Unlike latent generative models that rely on a separate decoder to map latent representations to signed distance fields, tri-vectors directly compute SDF values of query points. This eliminates the dependency on decoders after generation.
3. The representation based on tri-vectors is extendable to textured shape reconstruction and deformable shape reconstruction.

**Weaknesses:**

1. Tri-vectors have been previously explored in TensoRF (Chen et al., 2022) and Strivec (Gao et al., 2023).
2. Since the tri-vector representation is more parameter-efficient than SDF-based approaches, the paper would benefit from comparisons with HyperDiffusion (Erkoc et al., ICCV 2023), which represents shapes using neural SDF parameters.
3. The current experiments focus only on single category shape generation. Demonstrating unconditional generation across multiple shape categories would make the method more compelling, as shown in S2VS (Zhang et al., 2023).

**Questions:**

1. would be better if the authors could report the performance of per-shape SDF representations and S2VS (Zhang et al., 2023) in Table 2.
2. how many shapes are used from the Thingi10K and ShapeNet datasets in Table 2 evaluations?

---

### Official Review · Reviewer_nFFR · 2025-10-27

**Soundness:** 3
**Presentation:** 3
**Contribution:** 3
**Rating:** 6
**Confidence:** 2

**Summary:**

This paper formalizes a CP decomposition as Tri-Vectors for representing and generating continuous SDFs. The method emphasizes parameter efficiency, resolution scalability, and potential extension to textured and deformable settings. Experiments demonstrate competitive reconstruction and generation results with extremely few parameters, and the visual examples are appealing.

**Strengths:**

1. The paper presents a clear and intuitive application of classical CP tensor decomposition for continuous SDF representation and compression.

2. The method is resolution-independent and scalable. By adjusting vector resolution and the number of components, the model smoothly transitions from coarse to fine representations

3. The experiments cover diverse scenarios, including reconstruction and generative tasks, and extend to textured and deformable objects.

4. Tri-Vectors achieve comparable or even superior results to baselines with significantly fewer parameters, supporting the claim of a parsimonious representation.

**Weaknesses:**

Baseline comparisons are not rigorously controlled. Some listed methods differ in design purpose (scene-specific vs. generative), yet the comparisons are not made under equivalent parameter, training, or resolution settings. Several important baselines (e.g., modern VAE-latent models with strong decoders) are missing, making it unclear whether improvements arise from the representation itself or other implementation differences.

1. Limited reproducibility and missing experimental details. Essential hyperparameters—such as learning rate schedules, optimization steps, implementation specifics of SiT, random seeds for masking, and the rationale for the extremely small regularization weights ($\lambda$L2, $\lambda$var)—are omitted. The use of six million sampling points is computationally expensive, but the paper does not analyze its sensitivity or discuss possible lower-cost alternatives.

2. Insufficient evaluation of generative quality and diversity. The generation results mainly rely on COV/MMD/1-NN metrics without assessing sample diversity, mode collapse, or generation stability. Qualitative comparisons are subjective and lack detailed error analysis (e.g., distortion types or failure statistics).

3. Lack of limitation analysis and failure cases. The paper does not examine the performance degradation on complex topologies, fine-grained geometries, or noisy inputs. The robustness under violation of the “approximately separable structure” assumption remains unclear, and the random masking experiments are not deeply analyzed.

**Questions:**

1. The proposed method relies on multi-view alignment and generative priors, but it is unclear how robust the approach is when the input views exhibit significant occlusion or illumination inconsistency. Could the authors clarify the method’s behavior under such challenging conditions?

2. The paper emphasizes reconstruction fidelity, but there is limited discussion on the computational cost. What is the inference time and memory consumption for a typical sample compared to baseline methods?

---

### Official Review · Reviewer_hDan · 2025-10-29

**Soundness:** 1
**Presentation:** 1
**Contribution:** 2
**Rating:** 2
**Confidence:** 3

**Summary:**

This paper proposes Tri-Vectors, a 3D shape representation based on CP decomposition, for improving the efficiency of 3D generative models. The authors show their model as a solution to the trade-offs faced by existing methods, such as neural implicit functions and tensor representations.

**Strengths:**

While the topic is interesting, and the authors tried to integrate classic mathematics into generative settings (which is an interesting attempt), but the paper suffers from weak presentation and argumentation.

**Weaknesses:**

The paper in its current form is dense and difficult to follow. The problem definition is unclear, and the motivation for using CP decomposition is weak. The paper lists common trade-offs of existing methods (like, heavy compute or scene-specific), but didnt provide details for why these are the most bottlenecks.

There is no description of CP decomposition in the introduction, and the motivation for using it as the solution for a generative model is not well-defined. In my opinion, simply saying that a technique "has not been explored" in a generative setting, is not a sufficient technical motivation. There is no argument for why CP decomposition is suited to overcoming the limitations of other generative representations. Indeed, throughout the paper (specially in the introduction), the "why" is missing.

The related work section is weak. It doesn't define the key contributions and limitations of prior works. For example, in section 2.2, the authors list some transformer and diffusion models. Then, the last sentence is about their Tri-Vectors. There is no connection between them. It doesn't explain why is Tri-Vectors better than a transformer-based model? or how it solve the high dimensionality problem of diffusion models in a new way?

Beyond the lack of clarity, i have a question about figure 2.  In part (a), what is optimization block? If this block shows the Tri-Vectors, so this is the most important part of the representation idea and must be clearly defined, or maybe with a formulation.

In part (b), the Tri-Vectors are fed into an encoding block before going into the SIT network. This is confusing, because the Tri-Vectors are already the shape encoding, so why are using encoding an encoding? maybe this needs some justification.

Another confusing point is l_2 loss. The figure shows the loss is calculated between the model output and a target as, x_0 - x_t.  Am I right? maybe the authors need to justify it and explain what advantage it provides over diffusion model losses? It is not clear what each block or step contributes to the model.

**Questions:**

My questions are placed in the weaknesses block.

---

### Official Review · Reviewer_g5fd · 2025-11-03

**Soundness:** 2
**Presentation:** 2
**Contribution:** 2
**Rating:** 2
**Confidence:** 5

**Summary:**

This paper represent 3D shapes with 3 orthogonal vectors. And run diffusion model to generate such 3D representation.

**Strengths:**

-

**Weaknesses:**

This paper is outdated. Similar to tri-planes, Tri-Vectors appear to be more compact and data-efficient. However, generating such data representations has proven difficult, as they collapse data along certain dimensions, leading to irreversible data ambiguities.
The research community has shifted from using tri-planes to Vecset and Trellis. This paper misses comparisons with these methods and only compares with methods from before 2024. It is recommended to make a major revision before resubmitting, including:
1) Comparison with more recent approaches like Vecset and Trellis.
2) Scaling up the method to verify its robustness across complex shape, and general categories.

**Questions:**

-

---

### Meta-Review · Area_Chair_8QQm · 2025-12-21

**Summary:**

The reviewers’ concerns mainly focus on the novelty, positioning, and clarity of the proposed Tri-Vectors representation. Multiple reviewers question whether the approach is outdated compared to recent representations and whether the paper sufficiently explains why CP decomposition is well-suited for modern generative modeling. Additional concerns include limited comparisons with recent methods, unclear motivation and presentation, and insufficient experimental analysis of scalability, diversity, and robustness.

**Reviewer Concerns:**

The authors did not submit a rebuttal.

**Reviewer Scores:**

There was no discussion between the authors and the reviewers.

---

### Decision · Program_Chairs · 2026-01-26

Reject